# Electrical Discharge Machining of Oxide Nanocomposite: Nanomodification of Surface and Subsurface Layers

**Sergey N. Grigoriev** [1], **Marina A. Volosova** [1], **Anna A. Okunkova** [1,*], **Sergey V. Fedorov** [1], **Khaled Hamdy** [1,2], **Pavel A. Podrabinnik** [1], **Petr M. Pivkin** [1], **Mikhail P. Kozochkin** [1] and **Artur N. Porvatov** [1]

[1]   Department of High-Efficiency Processing Technologies, Moscow State University of Technology STANKIN, Vadkovsky per. 1, 127055 Moscow, Russia; s.grigoriev@stankin.ru (S.N.G.); m.volosova@stankin.ru (M.A.V.); sv.fedorov@icloud.com (S.V.F.); eng_khaled2222@mu.edu.eg (K.H.); p.podrabinnik@stankin.ru (P.A.P.); p.pivkin@stankin.ru (P.M.P.); astra-mp@yandex.ru (M.P.K.); porvatov_artur@mail.ru (A.N.P.)

[2]   Production Engineering and Mechanical Design Department, Faculty of Engineering, Minia University, Minia 61519, Egypt

*    Correspondence: a.okunkova@stankin.ru; Tel.: +7-909-913-1207

**Abstract:** The work is devoted to the research of the changes that occur in the subsurface layer of the workpiece during electrical discharge machining of conductive nanocomposite based on alumina with the use of a brass tool. The nanocomposite of $Al_2O_3$ + 30% of TiC was electroerosively machined in a water and hydrocarbon oil. The process of electrical discharge machining is accompanied by oscillations that were registered by diagnostic means. The obtained surface of the samples was researched by the means of scanning electron microscopy and X-ray photoelectron spectroscopy. The observed surface and subsurface changes provide grounding for the conclusions on the nature of processes and reactions that occur between two electrodes and nanomodification of the obtained surfaces that can be an advantage for a series of applications.

**Keywords:** electrical erosion; ceramic nanocomposite; sublimation; chemical reactions; submicrostructure; removal mechanism; fracture; erosion wear; monitoring; acoustic emission

## 1. Introduction

Electrical discharge machining is a well-known technology used for processing conductive materials [1–4], while there are always developed techniques to subject the non-conductive material to electroerosive wear [5–8]. However, these techniques do not show any outstanding results due to small volumetric performance and expensive assisting materials—nanoparticles of precious or rare materials, labor-intensive techniques of conductive layers applications, etc. Moreover, the character of the processes often is not stable and can hardly be called electroerosive, as some chemical destruction of the materials is more likely to occur than erosive destruction under electro current [9,10].

The idea of strength of ceramic composites and nanocomposites obtained by advanced sintering techniques with the conductive particles or nanoparticles has been known from the middle of the 1980s [11–14], and by the end of the 1990s, works devoted to the real experimental results appeared [15–17]. The current state of the mentioned research subject is still far from the industrial application, as the processed workpiece with a fine surface quality was never presented. This could be related to the low understanding of the nature of that which occurs between electrodes.

The electrical discharge machining is based on the principals of electrical erosion of the materials under current pulses [18–20]. Two electrodes are placed on the distance that correlates to the

conductivity of the material to be processed and there is dielectric permittivity of the medium between two electrodes. The material with higher conductivity requires a larger space between electrodes [21–23]. The material with lower conductivity is normally called the workpiece electrode, while material of high conductivity is a tool electrode—the destruction of the workpiece should occur more intensively that the destruction of the tool, otherwise the technology has no sense.

During the development of the technology of electrical discharge machining of the new ceramic composites and nanocomposites, many factors should be taken into account.

Firstly, these materials have low conductivity and require a minimal possible discharge gap (which, by precounting, should be in the range of 5–6 μm for some materials) that hampers evacuation of the erosion products from the working zone [24]. Thus, some measures should be taken to provide replicability of the dielectric in the discharge gap—intensive dielectric flows under pressure or other technological intensification measures, for example, ultrasound emission, which can assist the destruction of the agglomerates of the erosion products and intensify the dielectric flows in the zone [25–27].

The low conductivity of the ceramic composites and nanocomposites leads also to the following problem of its intensification and hampers any attempts to improve it. Many authors propose using measures on it that are related to the assisting of the machining by conductive powders or layers, which should introduce conductive particles to stimulate pulses on it, instead of the workpiece [28–30]. However, it shows its effectiveness only in micro-electrical discharge machining, when evenly weighted nanoparticles of silver or carbon nanotubes are uninterruptedly delivered in the interelectrode gap under pressure [31,32]. As it can be seen, the application of the developed measures hampers the effective evacuation of erosion products. Thus, the developed approach is still far from industrial application.

The next problem is also related to the low conductivity. If we try to machine ceramic composites and nanocomposites with a wire tool electrode, then we meet the problem of its self-oscillation. During machining, the conventional conductive materials and tools vibration are never a problem as the amplitude of the wire is rather less than the discharge gap and can be evaluated in a few nanometers at the distance between the nozzles of less than 100 mm [33]. During machining with the materials that have low conductivity with a smaller discharge gap and agglomeration of the erosion and assisting products in the working zone, short circuits caused by many factors including wire self-oscillations can occur even at a distance less than 100 mm [34,35].

One more problem is related to the low conductivity of the materials to be processed hampering electrical discharge machining. The control system of the machine uses the system of electrical pulse monitoring. Then, it counts all occurred pulses in the interelectrode gap. However, only a part of pulses can be called working and a part of them are idle. The proportion of working pulses by all of them can be called, in these conditions, effectiveness of the processing [23,33]. The process can be called effective for different conditions when this ratio is above 70% or even close to 100%. The idle pulses are always addressed to the destruction of erosion products when the working pulses are addressed to the destruction of the workpiece electrode. One of the solutions can be in the development of the multiparameter control system that counts not only every occurred impulse in the interelectrode gap, but only working pulses, using a monitoring system based on acoustic emission [36,37]. The actuality of the monitoring system's development, based on acoustic emission, that occurs at the moment of workpiece destruction, can be even higher in the context of innovative electrical discharge machining assisted with conductive powders or pellets or nanotubes.

Another point is related to the temperatures achieved in the working zone. As it is known, ceramics are heat-resistant materials that require high temperature in the working zone that should be achieved to make all components of the material sublimated. The problem is in fact, that at the moment when these temperatures will be achieved, the easy-to-melt conductive material of the tool will be already sublimated and the pulse will be interrupted with the consequent cooling of the working zone. It is obvious that the next pulse will not achieve any progress by reaching the same temperatures. Even uninterrupted renewing and feeding of the tool in the working zone can lead only to the large

consumption of the tool material or even to the short circuit but not to the stable and effective cutting process [38–41].

The next point is related to the nature of the destruction that occurs at the electrodes. The most advanced research showed success in processing zirconia and, based on it, composite and nanocomposites, when electrical discharge machining of any composite based on alumina, or alumina itself, shows such a low quality of machined surface that it is difficult to call the precision process. However, most of the authors cannot explain why it is so different and refer to conclusions on pyrolitic carbon and melting [5,32,42]. As we know, the carbon cannot be liquid or melted at normal pressure [43–46] and the temperatures that reach in the discharge gap are close to 10,000 °C, which correlates to the state of the substance called low-temperature plasma. In the conditions of the formed discharge channel between electrodes, the material of its surfaces sublimates. In other words, it changes its state from solid to low-temperature plasma bypassing the liquid phase. Only by understanding this obvious nature of the material destruction, can the destruction of the carbon material and its active use in electrical discharge machining in the role of tool electrodes and assisting powder be explained. However, sublimated materials interact between each other and sublimated components of working medium with the formation the substances that settle on the machined surfaces. In the case of machining zirconia in hydrocarbonates, it is a conductive zirconium carbide that assists further processing; in the case of machining alumina in hydrocarbonates, the nonconductive and pyrotechnically active aluminum carbide that is often called "pyrolitic carbon" or "PyC" (this is, however, not quite an exact term from a scientific point of view) is formed [47–50].

Moreover, traditionally, manufacturers of the equipment firmly recommend not using oil for machining aluminum-containing materials since non-conductive ionic bonded salt carbide (acetylide) is formed, where the bond's ionic nature leads to a high melting point and solubility in water and acids. Aluminum carbide forms at 1100 °C; it is abundant black sediment that can be observed during machining. It can explosively react with water (×12) and with methane escape (×3): 144 g of $Al_4C_3$ produces $3 \times 22.4$ L of $CH_4$. Adsorption of $Al_4C_3$ by machine filtration system can damage it. Aluminum carbide decomposes at 2200 °C to metallic aluminum with hydrogen (×6) and accompanied by methane escape (×3). It reacts with oxygen (×6) and with the formation of alumina and carbon dioxide (×3) at 650–700 °C due to its higher electronegativity ($\chi = 8.11$ for oxygen and 5.19 for carbon). It should be noted that all acetylides are not stable at normal conditions and can explode even with a tiny move ($Au_2C_2$, the best known as initiating explosives are silver and copper acetylides—$Ag_2C_2$, $Cu_2C_2$), and they exhibit dielectric properties [8].

However, the researchers continue to use oil or other hydrocarbons (kerosene) for unsuccessful attempts to machine alumina and alumina-based materials and hesitate to explain why it does not work when the surfaces' quality is under the required level.

Furthermore, as it was mentioned before, the occurred physical sublimation of the electrodes and chemical interaction of the components occurs faster for easy-to-melt materials. During the electrical pulses' interruption, the formed substances depose on the formed nanocarcass or nanoframe of the heat-resistant components of the workpiece material [51–55].

The purpose of the study is the research of physical phenomena of electrical discharge machining of oxide ceramic nanocomposite, and the classification of the nature of the observed defects.

This study aims to research the nature of nanocomposite destruction under discharge pulses, the nanomodification of the machined surface that occurs by means of scanning electron microscopy, to analytically research the chemical interaction of the workpiece and working medium components, and classify the obtained defects of the ceramic nanocomposite (Figure 1).

The scientific novelty of the work is new data on physical phenomena that occurred between the tool and workpiece electrode depending on the working medium in the case of ceramic nanocomposite machining, the classification of the obtained machined surface defects, and new data on nanomodification of the machined surfaces and chemical interactions of the near-surface layer after processing.

The tasks of the study are: adaptation and testing of the vibroacoustic monitoring mean for the machining of nanocomposites; research on the morphology and chemical content of the subsurface layer of machined surfaces; classification of the observed defects and traces of destruction; research on the medium content's influence on the composition of the subsurface layer after the machining of nanocomposite–nanomodifed conductive alumina $Al_2O_3$ + 30% of TiC.

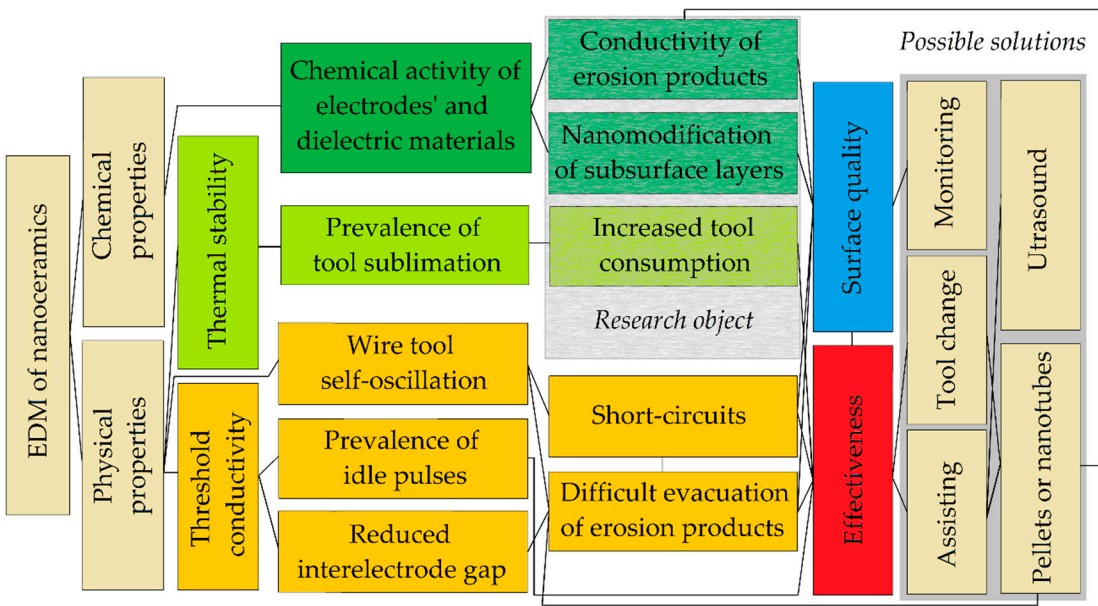

**Figure 1.** Current state of the obstacles that have occurred, their influence on the machinability of new materials and the quality of the obtained surfaces, the main trends in searching for solutions in the electrical discharge machining of ceramic composites and nanocomposites.

## 2. Materials and Methods

### 2.1. Materials and Discharge Gap

The nanocomposite samples were sintered of $Al_2O_3$ powder (Alcoa, New York, NY, USA) with 30% of TiC nanoparticles (Plasmotherm, Moscow, Russia) at 1450 °C in the graphite dies, with a heating rate of 100 °C·min$^{-1}$, under a uniaxial pressure of 80 MPa [56–58] by a hybrid current-assisted spark plasma sintering furnace KCE direct / hybrid heated 25-SD (FCT Electronic GmbH, Munich, Germany). The average particle diameter of powder granules was of 0.53 µm for $Al_2O_3$, and 0.6 µm for TiC.

Powders were preliminarily mixed in a multidirectional Turbula shaker mixer (Eskens B.V., Alphen aan den Rijn, The Netherlands) in ethanol at 150 rpm for 24 h. Lyophilizer FreeZone2.5 (LabConco, Kansas, MO, USA) was used for drying of the slurries to avoid TiC agglomerates in the ceramic matrix and additional sieving [7,8]. The collector temperature was of 50 ± 2 °C, the shell temperature was of +23 ± 2 °C; the chamber pressure was of 0.02 ± 0.01 mbar.

The control sample was made of ceramic composite VOK-6O (analogue of K01 by ISO, $Al_2O_3$—70%; TiC—30%) [59–61]. The sample thickness is of 10 mm.

An eddy current conductivity meter Sigmascope SMP10 (Helmut Fischer GmbH, Sindelfingen, Germany) measured the specific electrical resistance of the materials ρ used in the experiments (Table 1, Figure 2a). The device measures the material electric conductance in Siemens (S) and the percentage of the electrical conductance of the control sample produced from annealed bronze in the range of 1 ÷ 112%. All measured values were converted to $\left[\frac{\Omega \cdot mm^2}{m}\right]$. It should be noted that the specific electrical conductivity of $Al_2O_3$ is $2.10 \times 10^5 \div 10^6 \frac{\Omega \cdot mm^2}{m}$ at +20 °C and ~10.0 $\frac{\Omega \cdot mm^2}{m}$ at +1000 °C, and ~$6.0 \times 10^3 \frac{\Omega \cdot mm^2}{m}$ for TiC.

The melting/sublimation points of the materials *T* are provided in Figure 2b [62–66]. The sublimation point of the nanocomposite is determined by the most low-melting component; it is alumina with the temperature *T* of 2072 °C in the case of the chosen nanocomposite.

**Table 1.** Specific electrical resistance ρ of some materials at +20 °C.

| Material | Specific Electrical Resistance ρ $\left[\frac{\Omega \cdot \text{mm}^2}{\text{m}}\right]$ |
| --- | --- |
| Brass alloy CuZn35 | 0.065 |
| $Al_2O_3$ + 30%TiC ceramic nanocomposite | 3.773 |
| VOK-6O ceramic composite (analogue of K01, standards of ISO) | 3.510 |
| Aluminum oxide [1] | $2.10 \times 10^5 \div 10^6$ |
| Titanium carbide [1] | $6.0 \times 10^3$ |

[1] Provided for references.

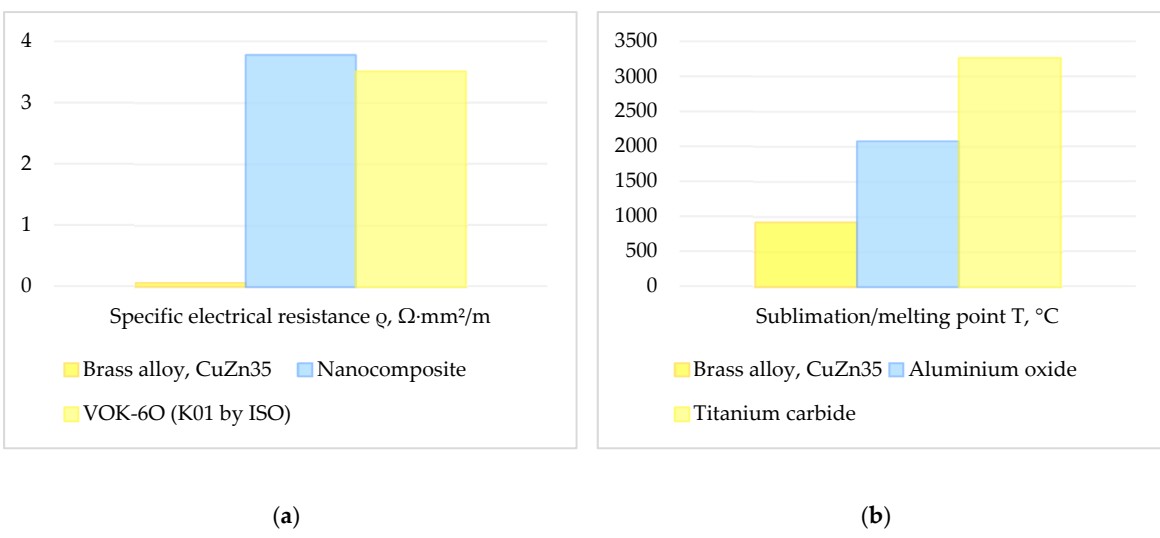

(**a**)　　　　　　　　　　　　　　　　　　　　　　　　　　(**b**)

**Figure 2.** Electrophysical properties of some materials: (**a**) Specific electrical resistance ρ at +20 °C; (**b**) Sublimation/melting point *T*.

The discharge gap $\Delta_{DB}$ is calculated by the next equations:

$$\Delta_{DB} = \frac{l - d_w}{2}, \tag{1}$$

$$\Delta_{DB} = \frac{l_m - l_p}{2}, \tag{2}$$

where *l* is the measured distance of the cut between two machined surfaces, mm; $d_w$ is the diameter of the wire tool, mm; $l_m$ is the measured distance between two adjacent cuts, mm; $l_p$ is the programmed distance between two adjacent cuts, mm.

## 2.2. Equipment and Methods

A four-axis wire electrical discharge machine AgieCharmilles CUT 1000 OilTech (GF Machining Solutions, Bern, Switzerland) was used for machining ceramic nanocomposite in Sorepi-LM oil medium—purified of sulfur, zinc and other components' mineral oil with refined paraffins (mixture of liquid hydrocarbons of $C_8$-$C_{15}$). A two-axis wire electrical discharge machine ARTA 123 Pro (OOO "Scientific Industrial Corporation "Delta-Test", Fryazino, Moscow Oblast, Russia) was used

for experiments with ceramic nanocomposites in water. The main characteristics of the machines are presented in Table 2.

**Table 2.** Main characteristics of wire electrical discharge machines used in experiments.

| Characteristic | Value and Description |
|---|---|
| *CUT 1000 OilTech* | |
| Max axis motions along the axes $X \times Y \times Z$, mm | $220 \times 160 \times 100$ |
| Max angle of conical machining, degree | $\pm 3°$ |
| Max weight of workpiece, kg | 35 |
| Accuracy of positioning along the axes, µm | $\pm 0.5$ |
| Achievable roughness $Ra$, µm | 0.05 |
| Dielectric medium | Mineral oil |
| Machine body | Solid |
| Max power consumption, kW | not confined |
| *ARTA 123 Pro* | |
| Max axis motions along the axes $X \times Y \times Z$, mm | $125 \times 200 \times 80$ |
| Max weight of workpiece, kg | not confined |
| Accuracy of positioning along the axes, µm | $\pm 1$ |
| Achievable roughness $Ra$, µm | 0.6 |
| Dielectric medium | Deionized water |
| Machine body | Solid |
| Max power consumption, kW | <6 |

The machines are located in a thermo-constant room to reduce the effect of ambient temperature on the results of processing. For all the experiments, a workpiece was immersed in a dielectric medium for 10 min before machining to avoid dimensional fluctuations associated with the difference in temperatures between the environment and working fluid. The dielectric height was established ~1–2 mm above the workpiece. The upper guide of the machine was placed at the minimum position 2–5 mm above the level of the dielectric [67,68].

The computer numerical control (CNC) programs were prepared manually, and the path offsets were not taken into account. The EDM-machine factors chosen following the recommendations [23,33,69–71] are presented in Table 3. The values are provided in equivalent units of the machine. The tool electrode used in all the experiments was a brass wire $d_w$ of 0.25 mm in diameter made of CuZn35 (Cu—65%; Zn—35%).

**Table 3.** Electrical discharge machining factors during experimental work for machining $Al_2O_3 + 30\%TiC$ in water.

| Factor | Value |
|---|---|
| Operational voltage, $V_o$ | 75, 85, 108 V |
| Pulse frequency, $f_p$ | 5 kHz |
| Pulse width, $T_p$ | 1 µs |
| Speed of the tool rewinding, $W_s$ | 7 m/min |
| Feed rate, $R_f$ | 0.4 mm/min |

*2.3. Monitoring*

Piezoelectric accelerometers were located at an upper wire guide and at a working table of the machine for vibroacoustic monitoring [23,33,35,69,72,73]. The signals received by the accelerometers were forwarded to preamplifiers, VShV003 amplifiers (Joint-stock company "Izmeritel", Taganrog, Russia), and an analog-to-digital converter E440 ("L-card", Saint-Petersburg, Russia), and were recorded at 1 min, 30 s, and 5 s before the end of processing with a personal computer (PC). Spectral analysis was performed at frequencies of 4–10 kHz. The measuring circuit is shown in Figure 3.

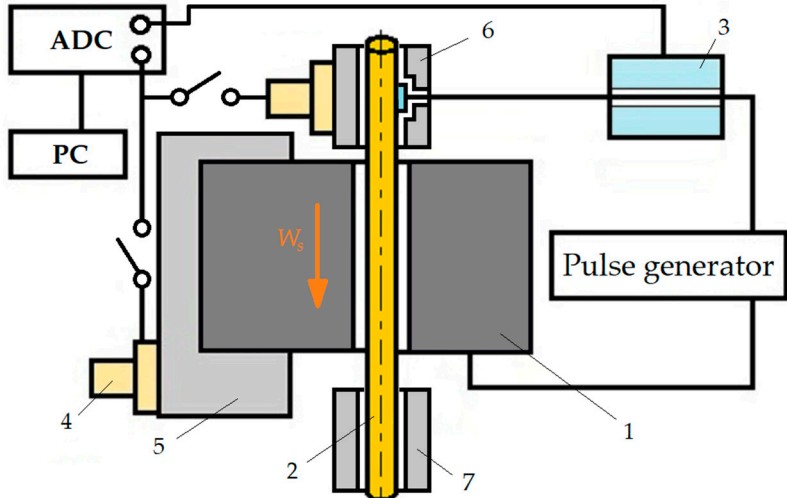

**Figure 3.** The layout of the sensors on the machine for electrical discharge machining of nanocomposites: (**1**) is a ceramic workpiece; (**2**) is a wire tool; (**3**) is a current sensor; (**4**) is accelerometers; (**5**) is a fastening system; (**6**) is an upper wire guide; (**7**) is a lower guide; ADC is the analog-digital converter; PC is the personal computer; $W_s$ is the direction of wire rewinding.

Pulse energy is the work done by the pulse in the interelectrode gap. It is inconvenient to use it in calculations, therefore, for independent generators, this energy is estimated by the average current [74]. In the experiments, a Hall sensor was used to record the discharge current. The effective values ($I$) of the signals from the Hall sensor after amplification and processing by a high-pass filter gave an idea of the energy entering the interelectrode gap. The signal depending on the energy used for material removal came from an accelerometer mounted on the elastic system of the machine (fastening system) from the side of the workpiece. The signal was preliminarily cleaned from low-frequency noise using a high-frequency filter, and then its effective amplitude was determined, the square of which is proportional to the energy of the signal arising in the elastic system under the influence of disturbing influences from the discharge pulses. The cutoff frequency of filters for vibroacoustic signals is 2 kHz, for current—100 Hz.

## 3. Results

### 3.1. Monitoring

Figure 4a shows the spectra of signals from the accelerometer and current sensor. As can be seen, the frequency of the operational discharges is marked $f_i$. It can easily be identified from the rest of the spectrum when the recorded signal amplitude shows disturbance in the interelectrode gap related to the destruction of the material in the wide frequency range. The changes in root-mean-square value (RMS) of the vibroacoustic signal recorded during processing give more adequate data about the state of machining (Figure 4b).

The shown example (Figure 4b) of parallel records of $A$ and $I$ after processing by high-pass filters. Graph 1 shows a gradual increase in $A$, finding it at a sufficiently high level for 12 s and falling to a minimum during the rest of the idle time of the wire tool. At the same time, $I$ rapidly increases at the beginning of the electrical discharge machining, remains at a stable level throughout the entire time of machining, but with further idle movement of the tool, $I$ does not fall (Graph 2), as can be seen in the example of $A$, and even increases in certain periods of time. This is because breakdown occurred through the medium- or short-term short circuits between the electrodes along the surfaces to be processed during this period. However, it is clear from the Graph 1 that no useful work is performed during this period.

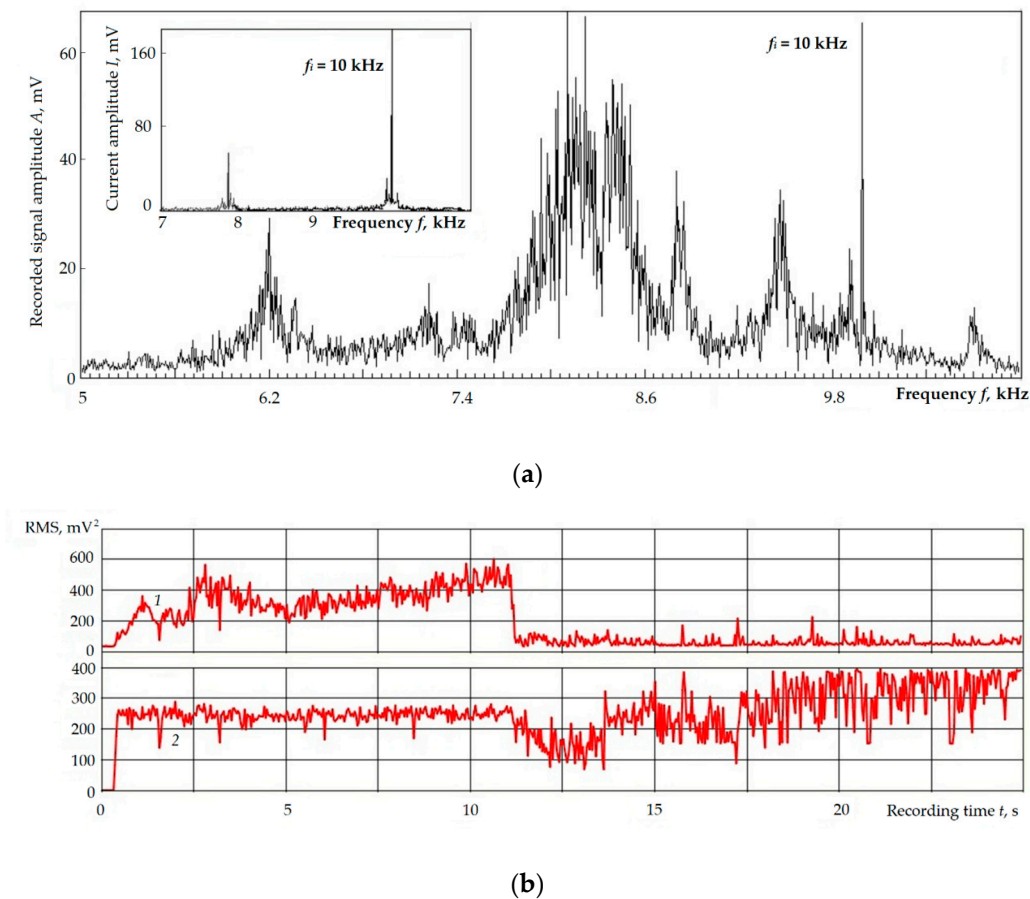

(a)

(b)

**Figure 4.** Spectrum of the recorded amplitude *A* and electrical current *I* signals and its root-mean-square value in recorded time: (**a**) spectra; (**b**) recorded root-mean-square value of the signal amplitude (**1**) and electrical current (**2**).

## 3.2. Roughness and Surface Topology

The research of the nanocomposite sample's microrelief after machining in oil and water medium is presented in Figure 5. The samples machined in hydrocarbon dielectric have a particular topology with the wire traces that correspond to the case of the heterogeneous workpiece structure or other difficulties in machining. At the same time, the topology of the surface after machining in water has a more even structure that is more similar to the surface profile after EDM and can be characterized as a typical "shagreen" type.

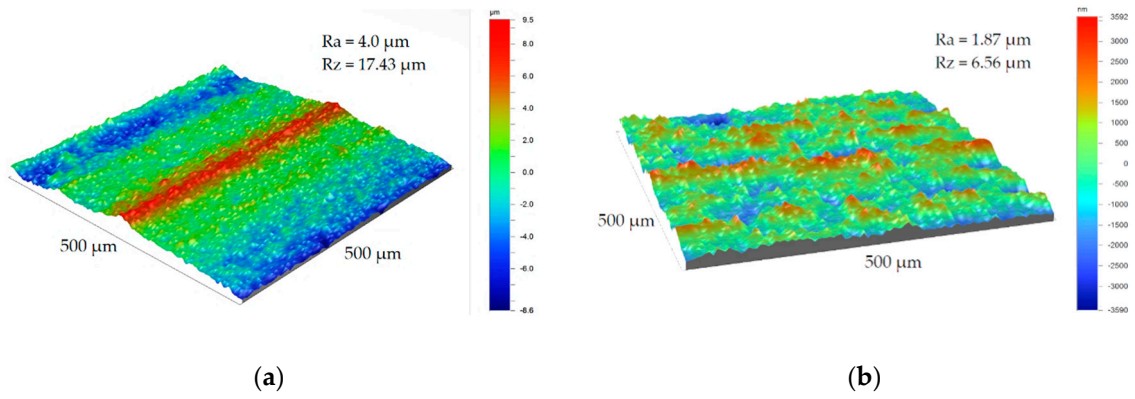

(a)                                                           (b)

**Figure 5.** Roughness of the $Al_2O_3$ + 30%TiC nanocomposite samples after machining: (**a**) oil; (**b**) water.

It can be related to the complex nature of material destruction during the EDM of alumina and alumina-based materials that has not only the presence of the thermal destruction under discharge pulses (EDM of nanocomposite in water) but also of the chemical interaction of workpiece electrode material with the medium (EDM of nanocomposite in oil). The formed dielectric sediment can change electrical conditions in the interelectrode gap and hampers the intensity of electrical discharges, leading to the reduction in the discharge gap that induces multiple short circuits that were observed during processing, with the formation of black dust clouds that settled in the tank.

The surface topology is presented in Figure 6. The visible pellets form the entire elaborate flakes in machining in oil and have a maximum size of $7 \times 5$ µm, with cracks and pores that are evidence of its secondary sublimation (Figure 6a). In machining in water, a set of pellets—flake—has a more interconnected character with a maximum size of $10 \times 5$ µm and with even more noticeable secondary sublimation evidence—cracks and pores (Figure 6b). The cracks are more evident for machining in water due to more intensive heat removal from the processing area that depends on the thermal diffusivity $\alpha$ of the used medium, which is $0.143 \times 10^{-6}$ m$^2 \cdot$s$^{-1}$ at +25 °C for water and $7.38 \times 10^{-8}$ m$^2 \cdot$s$^{-1}$ at +100 °C (similar to wood) for mineral oil.

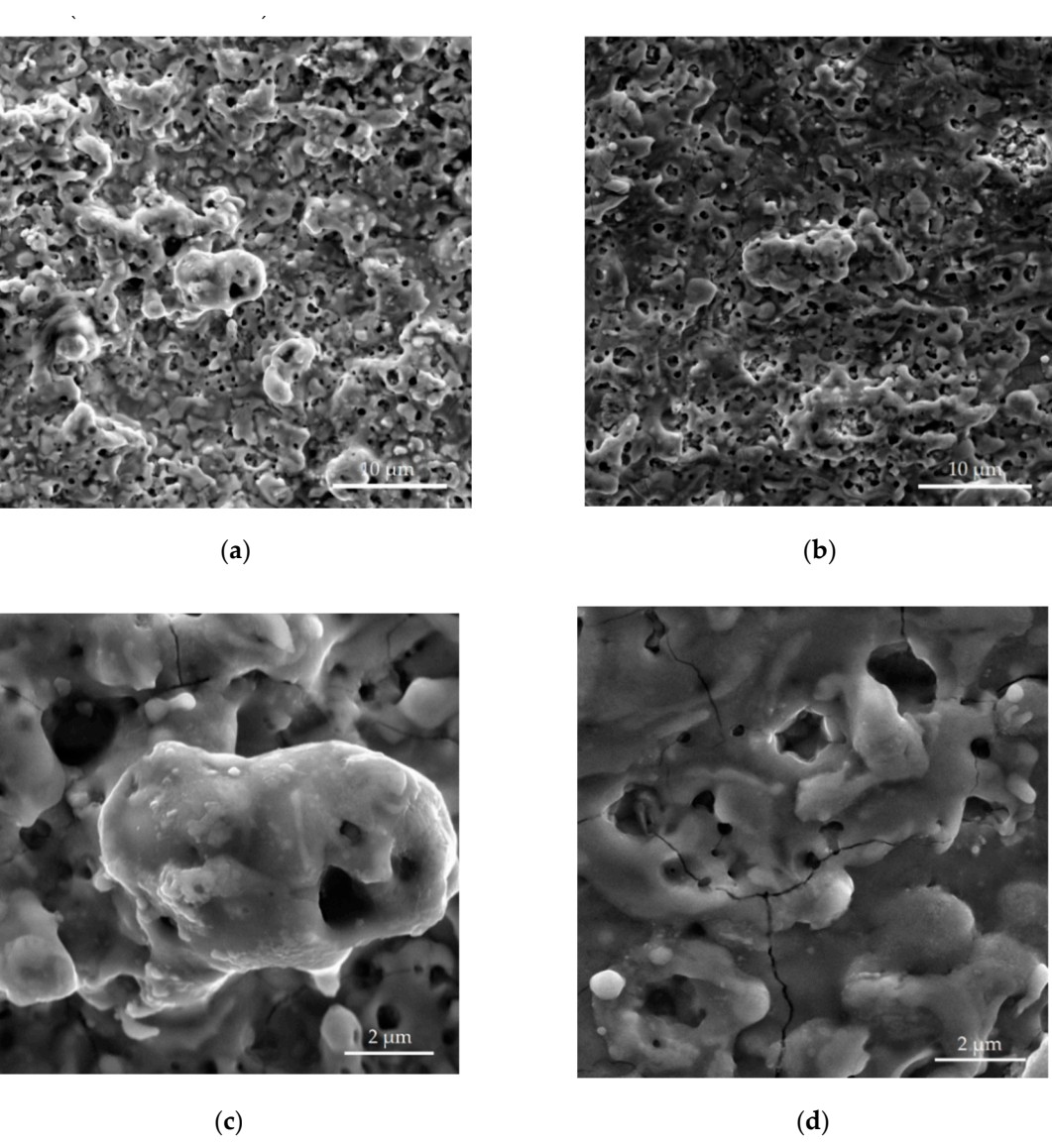

**Figure 6.** Topology of the Al$_2$O$_3$ + 30%TiC nanocomposite machined surface: (**a**) oil, 10.0 k×; (**b**) water, 10.0 k×; (**c**) oil, 40.0 k×; (**d**) water, 40.0 k×.

### 3.3. Chemical Content of Eroded Surfaces under Discharge Pulses

Figures 7 and 8 show the elemental distribution at the $Al_2O_3$ + 30%TiC nanocomposite machined surface with a magnification of 6.0 k×. The provided images show an even distribution of the chemical elements such as aluminum, oxygen and titanium on the surface machined in water (Figure 8a–c) when machining in the oil medium gives more local agglomeration of the chemical elements (Figure 7a–c). This can correlate to the more elaborate and pronounced topology of the surface produced in oil. Carbon has a specific distribution in machining in hydrocarbon oil with the formation of more saturated and unsaturated areas that do not correlate with the obtained topology (Figure 7d); machining in water provides an even carbon distribution over the entire observed area (Figure 8d). It should be noted that all colors for chemical elements at microphotographs and graphs are generated automatically by used software equipment.

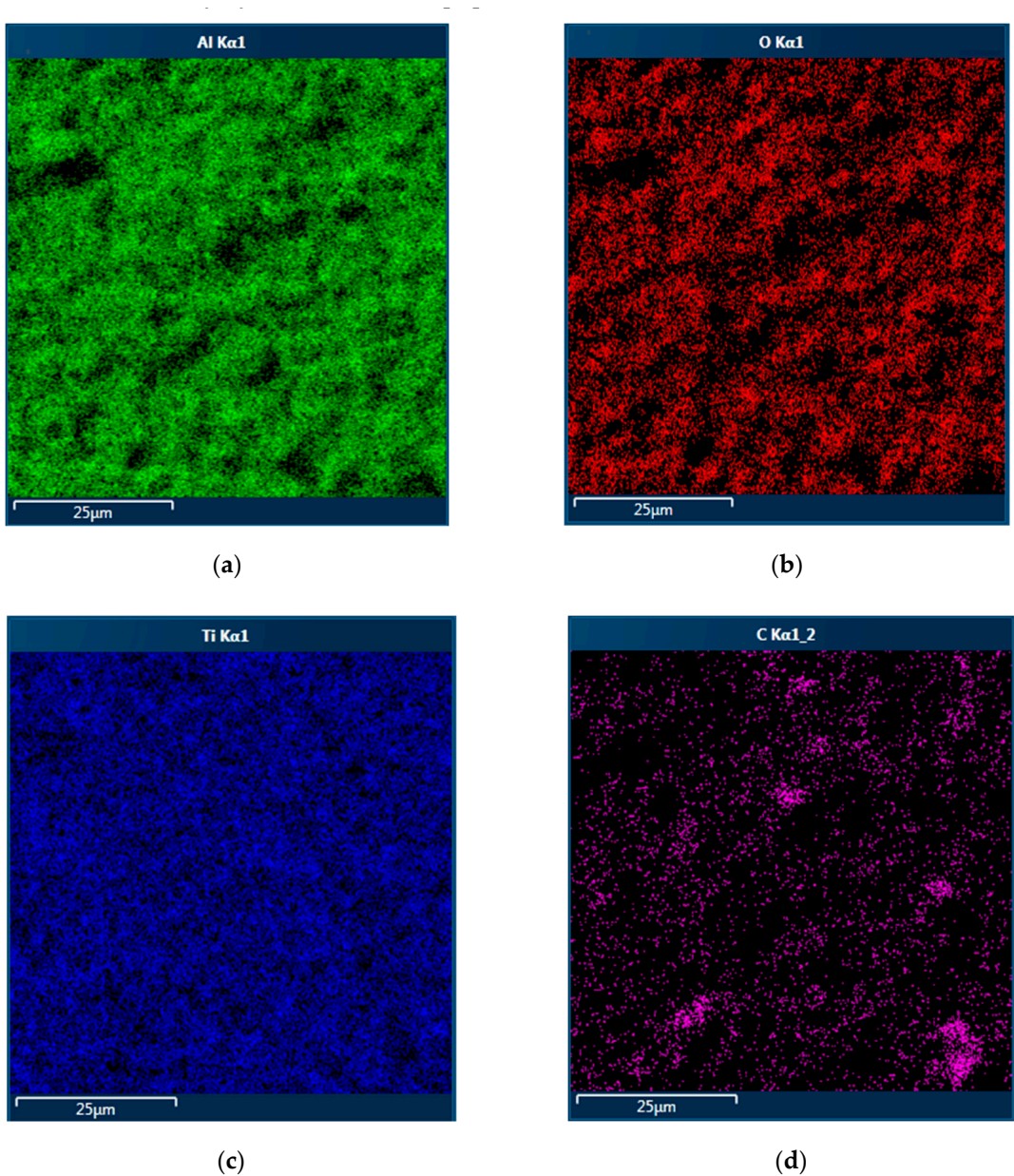

(a)   (b)

(c)   (d)

**Figure 7.** Elemental mapping of the $Al_2O_3$ + 30%TiC nanocomposite machined surface in oil medium: (**a**) Aluminum; (**b**) Oxygen; (**c**) Titanium; (**d**) Carbon.

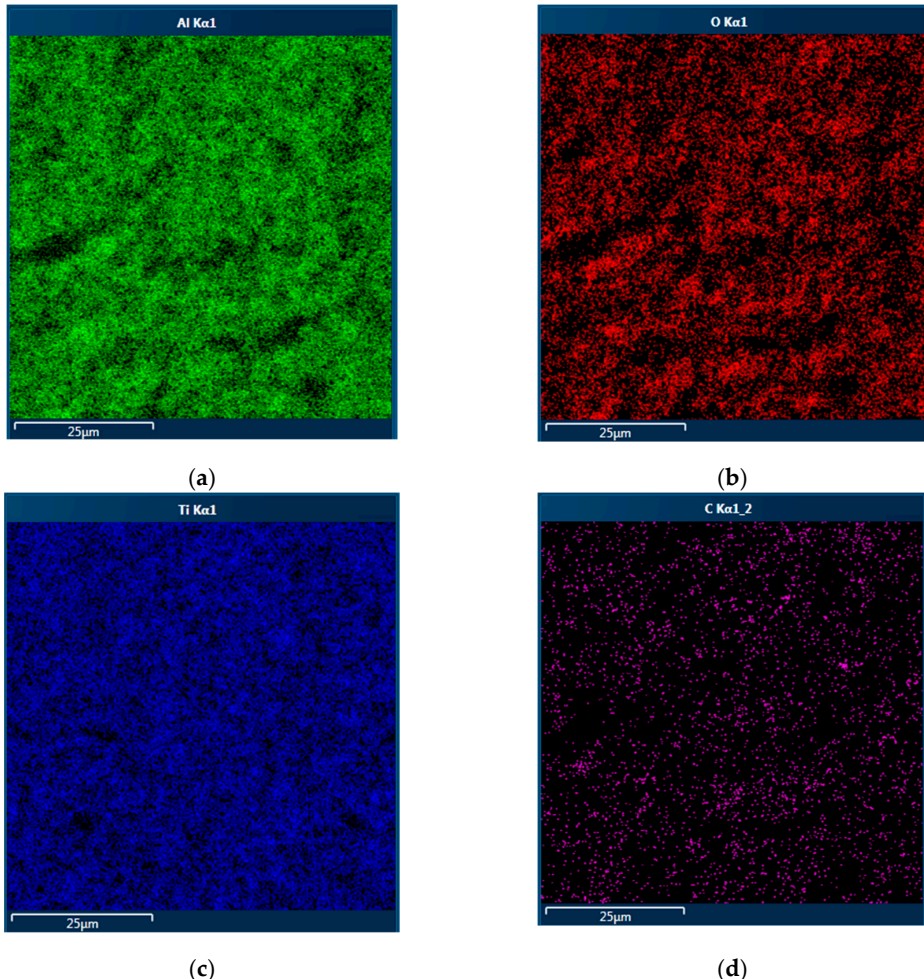

**Figure 8.** Elemental mapping of the $Al_2O_3 + 30\%TiC$ nanocomposite machined surface in water medium: (**a**) Aluminum; (**b**) Oxygen; (**c**) Titanium; (**d**) Carbon.

*3.4. Microstructure and Chemical Content of Cross-Section*

The microstructure and chemical analyses of the samples machined in oil are presented in Figure 6. The pictures show an even distribution of the nanocomposite components when the surface and subsurface layers demonstrate changes in the chemical content (Figure 9a). The distribution of aluminum and oxygen can demonstrate a denser presence of alumina (Figure 9b,c) in the surface layer and depletion of the titanium carbide (Figure 9d) despite the higher sublimation point $T$ of the TiC (3257 °C) that can be related to the high chemical reactivity of aluminum towards oxygen [75–77]. However, the subsurface layer saturated with carbon of oil medium is detected on the chemical map of the observed area (Figure 9e).

The microstructure and chemical analyses of the samples machined in water are presented in Figure 10. The similar chemical content of the surface and subsurface layer can be observed. However, the distribution of aluminum and oxygen has a more pronounced character and forms plaques (flakes) of secondary structures of the second order (Figure 10b,c) [78,79]. The titanium and carbon depletion can be also observed (Figure 10d,e) as the presence of a solid solution of aluminum and titanium and more complex compounds in the form of pellets (Figure 10b,d). The subsurface layer has also detected the presence of carbon but this sublayer is less pronounced and can be explained only by thermal influence (heat-affected zone), decomposition of titanium carbide and titanium depletion (Figure 10e).

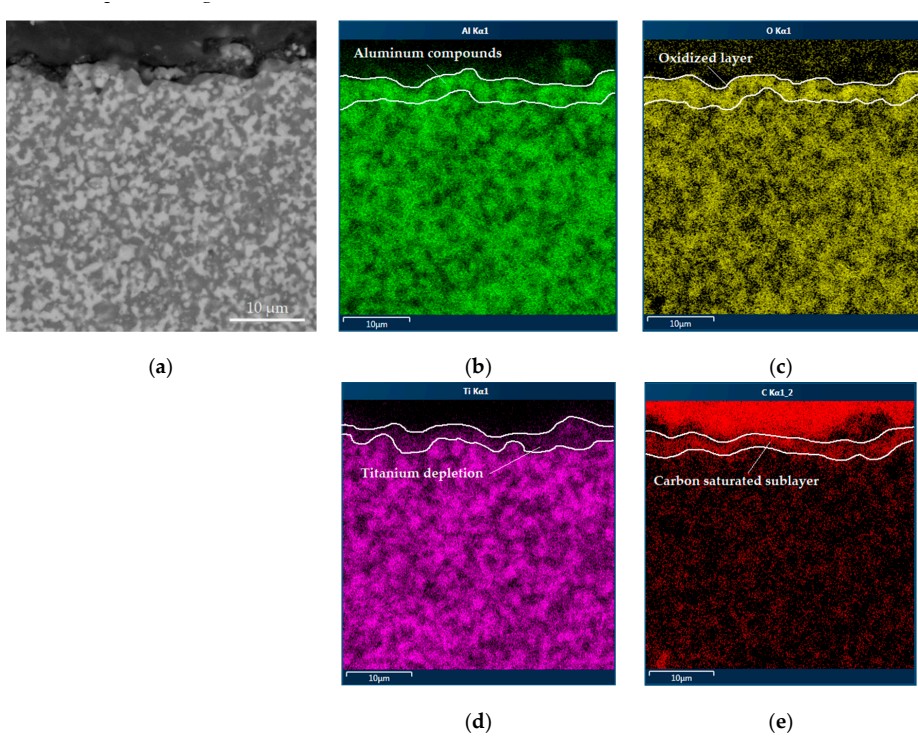

**Figure 9.** Microstructure and elemental mapping of the Al$_2$O$_3$ + 30%TiC nanocomposite machined in oil medium: (**a**) Microstructure in back-scattered electrons (BSE), 5.0 k×; (**b**) Aluminum; (**c**) Oxygen; (**d**) Titanium; (**e**) Carbon.

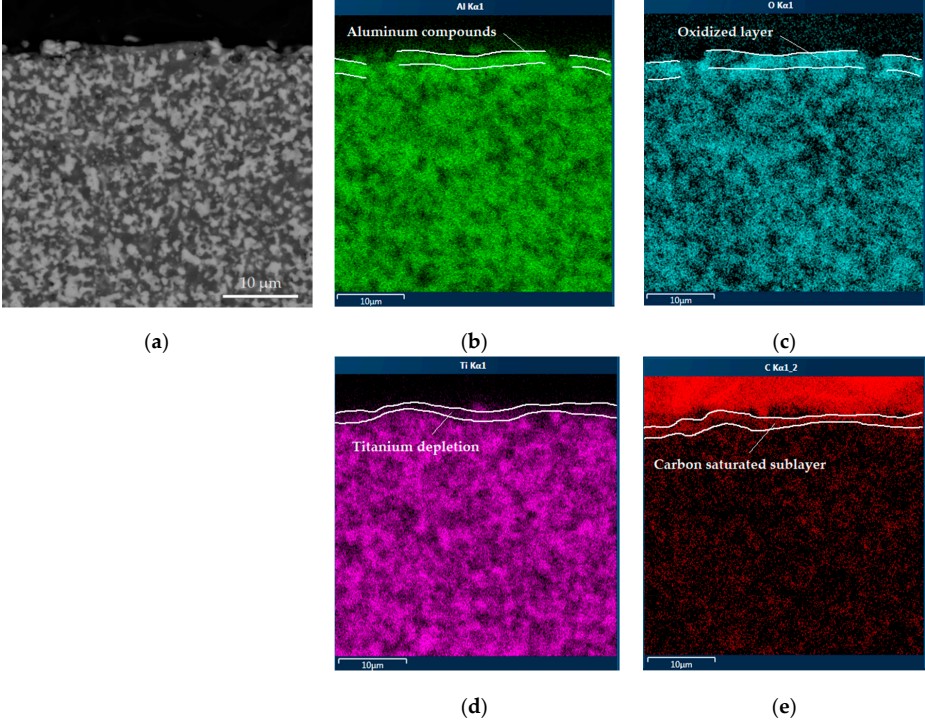

**Figure 10.** Microstructure and elemental mapping of the Al$_2$O$_3$ + 30%TiC nanocomposite machined in water medium: (**a**) Microstructure in back-scattered electrons (BSE), 5.0 k×; (**b**) Aluminum; (**c**) Oxygen; (**d**) Titanium; (**e**) Carbon.

The details of chemical analyses with higher resolution and magnification along the line (14 μm) of the sample machined in oil and water show a similar picture of the changes in the chemical content of surface and subsurface layers (Figure 11). Both samples demonstrate a more pronounced and deep saturation of the samples with aluminum and oxygen and a less pronounced spike in carbon content when titanium content gradually decreases towards the machined surface. The surface and subsurface layers demonstrate almost similar content—the sample machined in oil is more saturated with aluminum (Figure 11b) and the absence of the wire tool's chemical components.

### 3.5. X-Ray Photoelectron Spectroscopy

X-ray photoelectron spectroscopy of the surface and subsurface layers is presented in Figure 12. Tables 4 and 5 quantitively show X-ray peaks [eV] and the atomic percentage (at.%) of one type of atom to the total number of atoms.

**Table 4.** X-ray peaks of the machined $Al_2O_3$ + 30%TiC nanocomposite in oil.

| Chemical Element | Binding Energy $E$ | Binding Energy Peak, eV | Atomic % |
|---|---|---|---|
| Carbon | C1s Graphite | 283.8 | 16.64 |
| | C1s C-C | 284.4 | 17.06 |
| | C1s Carbide | 283.3 | 12.66 |
| | C1s C=O | 287.4 | 8.49 |
| | C1s C-O | 285.1 | 19.68 |
| Aluminum | Al2p3 Oxide | 74.3 | 2.75 |
| Nitrogen | N1s C-N | 399.3 | 3.61 |
| | N1s N-Si, Me-N | 397.6 | 0.23 |
| Calcium | Ca2p3 Carbonate | 346.8 | 1.58 |
| Oxygen | O1s C=O | 532.5 | 5.16 |
| | O1s Scan B | 530.9 | 6.55 |
| | O1s C-O | 531.7 | 3.66 |
| Copper | Cu2p3 Me | 932.4 | 0.16 |
| Zinc | Zn2p3 ZnO | 1021.7 | 0.11 |
| Silicon | Si2p3 organic, Si3N4, Silicates | 101.8 | 1.39 |
| Titanium | Ti2p3 TiC | 454.4 | 0.09 |
| | Ti2p3 TiO2 | 458.3 | 0.15 |
| | Ti2p3 TiN | 455 | 0.02 |

**Table 5.** X-ray peaks of the machined $Al_2O_3$ + 30%TiC nanocomposite in water.

| Chemical Element | Binding Energy $E$ | Binding energy peak, eV | Atomic, % |
|---|---|---|---|
| Carbon | C1s C-O | 285.5 | 30.54 |
| | C1s C=O | 288.3 | 4.85 |
| | C1s C-C | 284.6 | 25.42 |
| Aluminum | Al2p3 Oxide | 73.8 | 10.47 |
| Nitrogen | N1s C-NH2 | 399.6 | 2.67 |
| Calcium | Ca2p3 CaCO3 | 347.4 | 1.18 |
| Oxygen | O1s C=O | 532.8 | 5.77 |
| | O1s C-O | 531.7 | 9.49 |
| | O1s MeO | 530.5 | 8.67 |
| Titanium | Ti2p3 Me | 453.6 | 0.34 |
| | Ti2p3 TiO2 | 457.8 | 0.22 |
| | Ti2p3 TiN | 455 | 0.17 |
| Zinc | Zn2p3 Me | 1022 | 0.11 |
| Copper | Cu2p3 CuO | 932.9 | 0.12 |

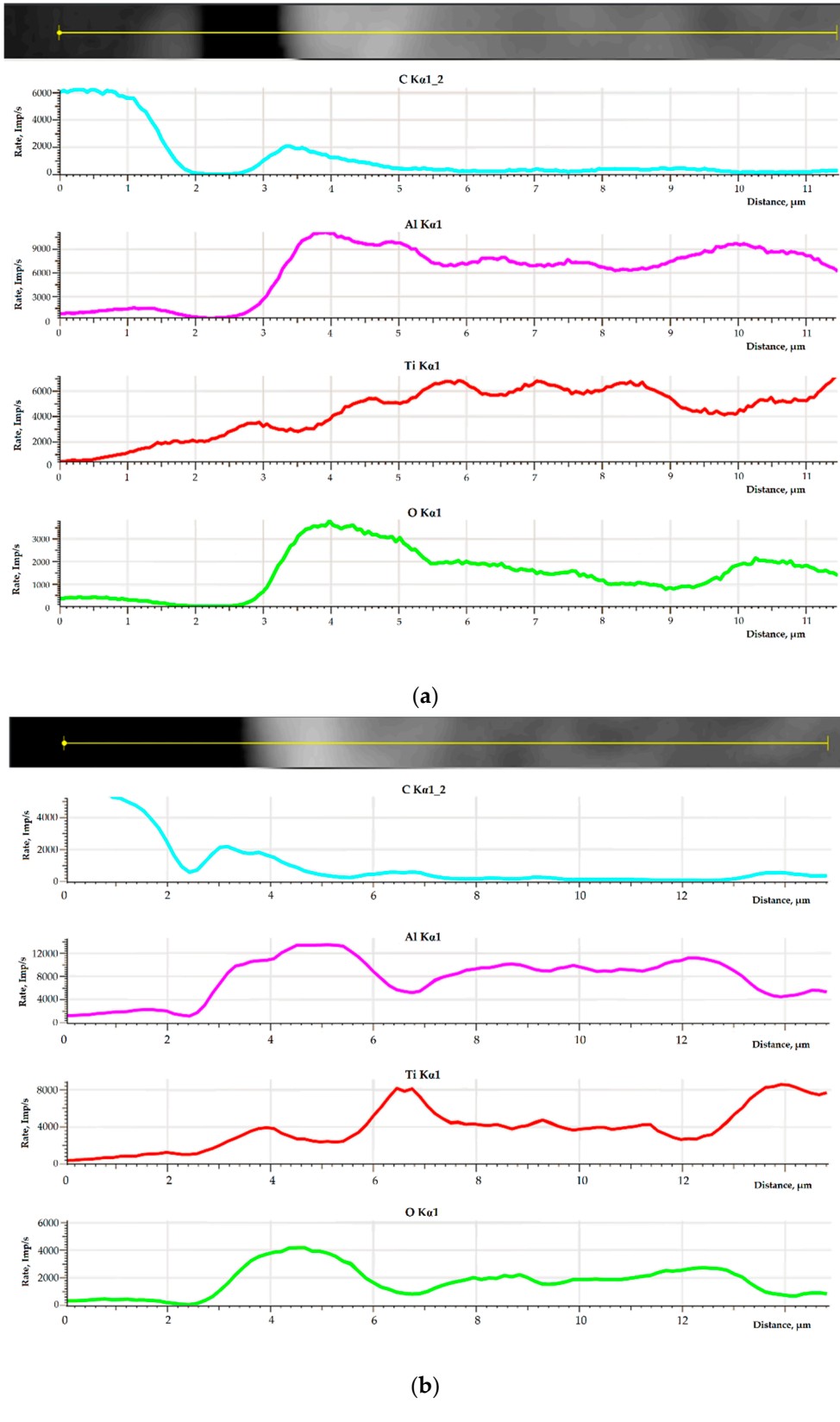

**Figure 11.** Chemical analyzes along the line of the $Al_2O_3$ + 30%TiC nanocomposite cross-section after machining: (**a**) Oil; (**b**) water, where a blue line is for carbon, a fuchsia line is for aluminum; a red line is for titanium, and a green line is for oxygen.

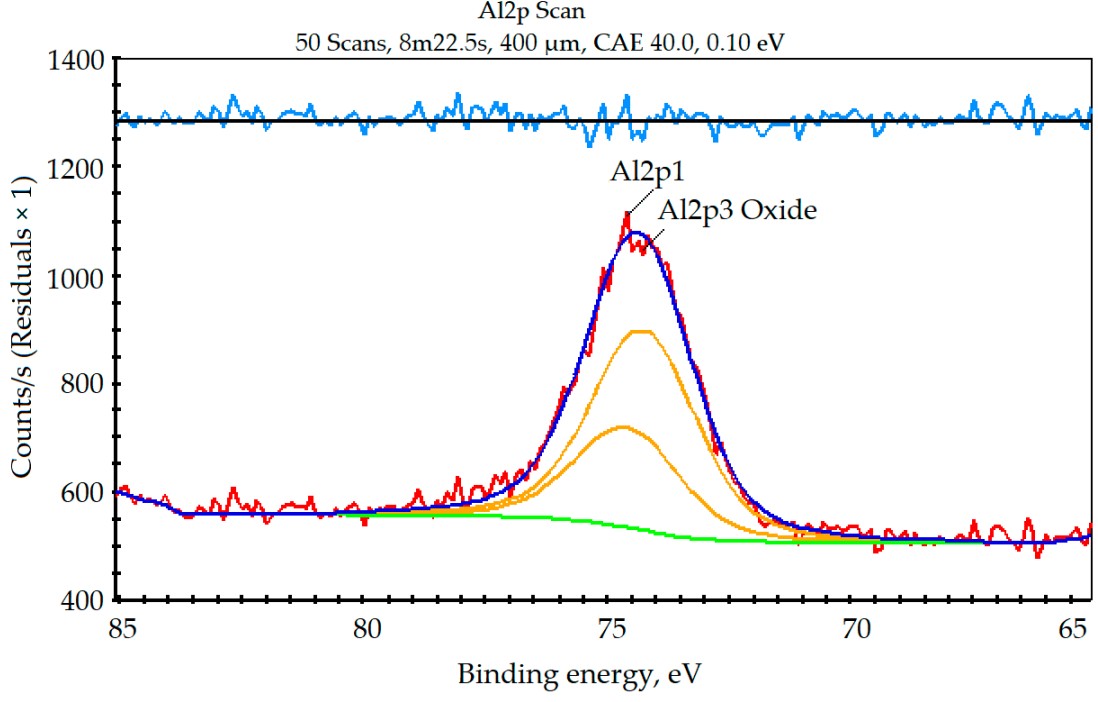

(**a**)

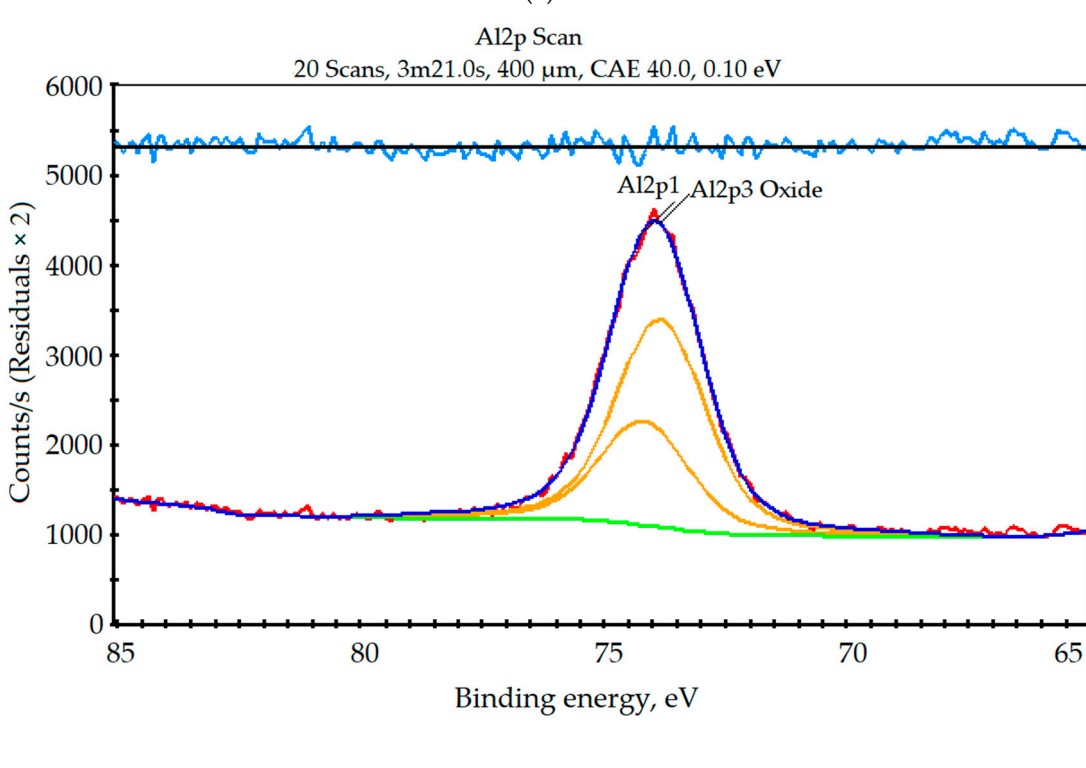

(**b**)

**Figure 12.** *Cont.*

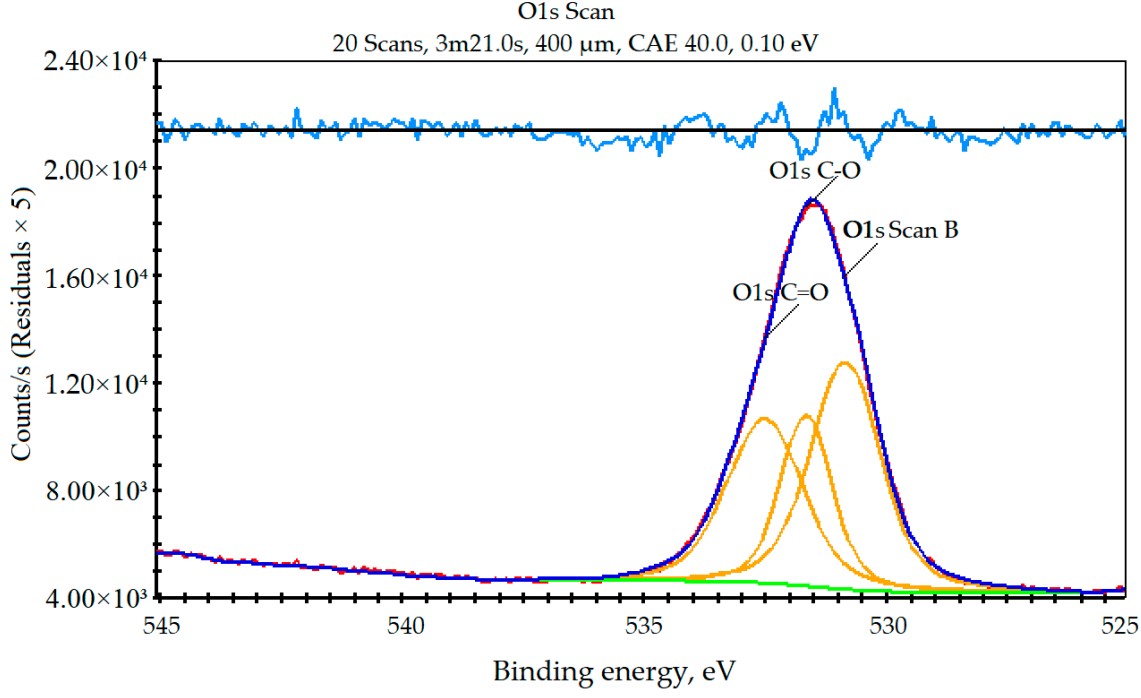

(**c**)

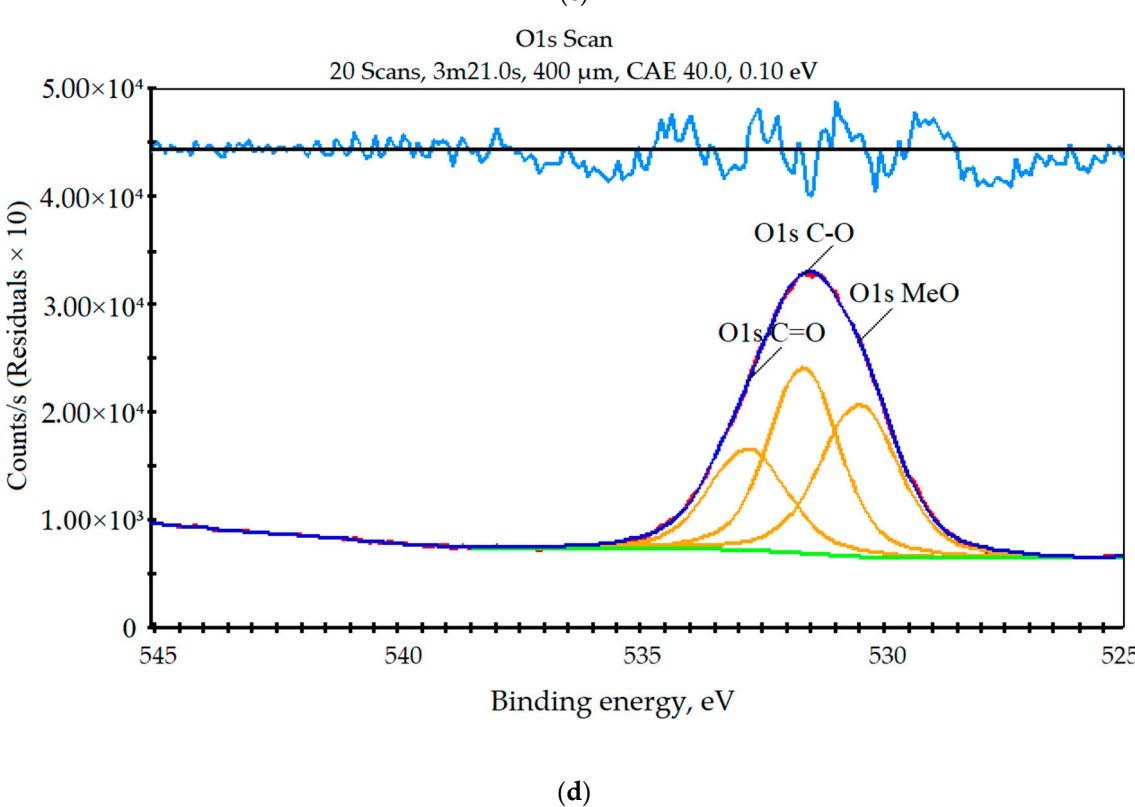

(**d**)

**Figure 12.** *Cont.*

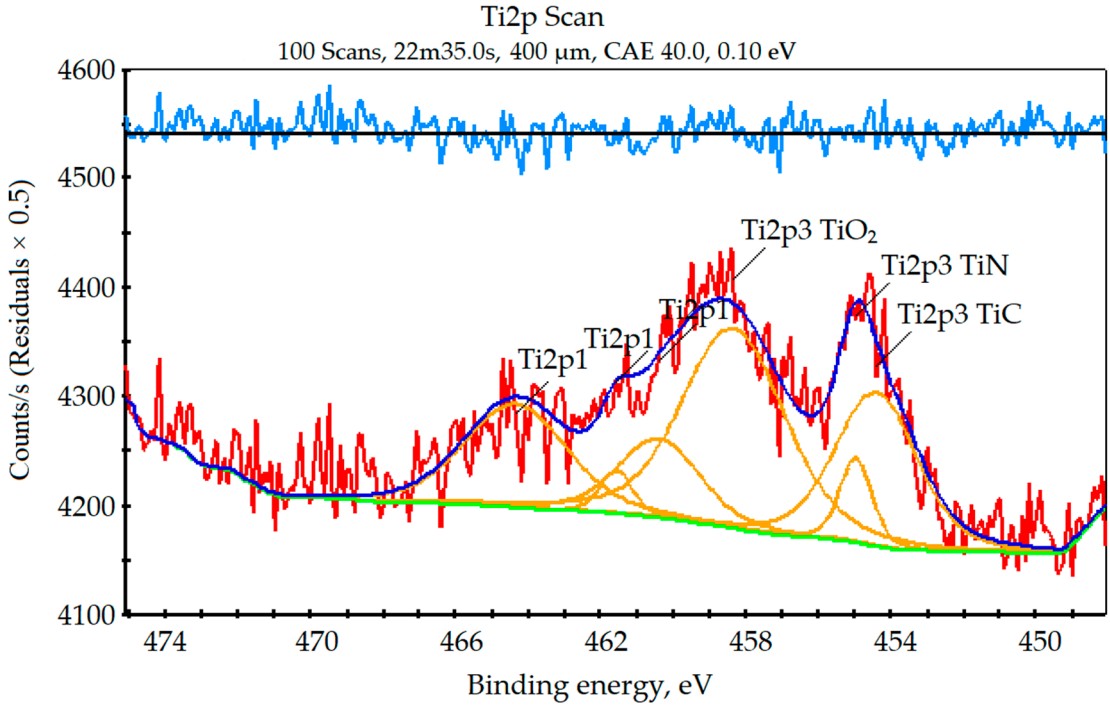

(**e**)

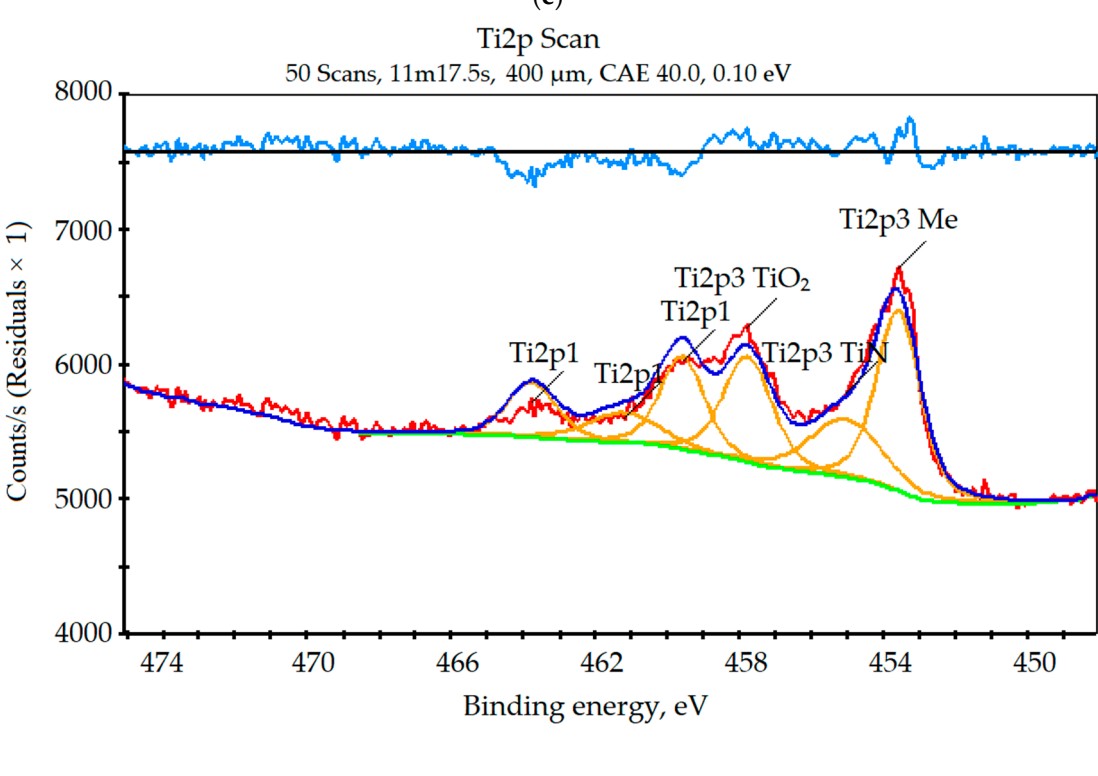

(**f**)

**Figure 12.** *Cont.*

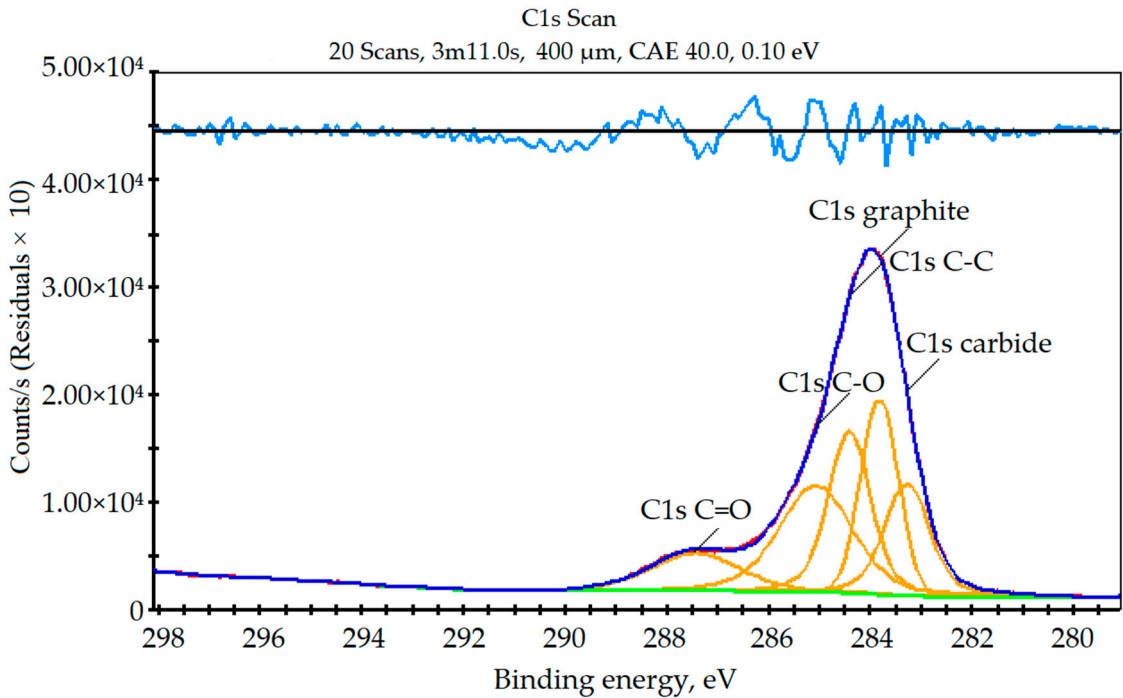

**(g)**

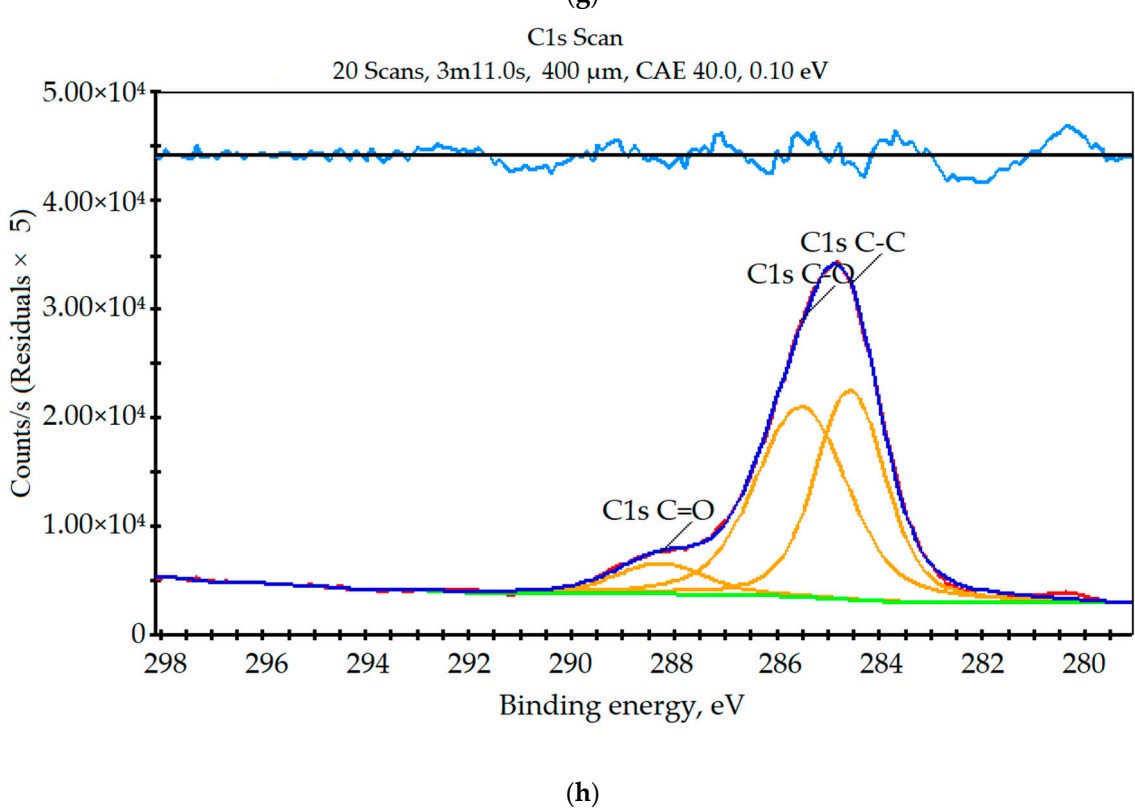

**(h)**

**Figure 12.** X-ray photoelectron spectroscopy of the $Al_2O_3$ + 30%TiC nanocomposite cross-section after machining: (**a**) Aluminum content, in oil; (**b**) Aluminum content, in water; (**c**) Oxygen content, in oil; (**d**) Oxygen content, in water; (**e**) Titanium content, in oil; (**f**) Titanium content, in water; (**g**) Carbon'content, in oil; (**h**) Carbon content, in water.

The graphs show a similar content of aluminum in the form of oxide. The sample machined in water has an alumina content by 3.8 times higher than the sample machined in oil, which can be explained by the active formation of $Al_3C_4$ in the form of sediment in the oil medium.

The sample machined in oil shows the bonding of titanium in carbides as well as the formation of $TiO_2$ and TiN ceramics, while the sample machined in water has a significant presence of metallic titanium and the same ceramics. However, the content of TiN in the sample machined in water is 8.5 times higher than for the sample machined in oil.

The presence of C-C, C-O, C=O bonds corresponds to the normal atmosphere contamination of the samples. Still, the summarized percentage of contaminations is lower by 1.35 times for the samples machined in oil. It can be explained by partial carbon bonding in the form of graphite and carbides.

Oxygen is bonded in the surface and subsurface layers in the forms that correspond to the mentioned atmospheric contaminations and, in the case of the sample machined in water, metallic oxides. The summarized presence of oxygen is 1.5 times higher for the samples produced in water than for the samples machined in oil.

It is quite interesting that both of the samples demonstrate the presence of the compounds that correspond to the chemical composition of the wire tool—brass. In the case of sample machined in oil, the presence of the metallic copper and zinc oxide was detected, while the sample produced in water showed an almost similar presence of metallic zinc and copper oxide.

The presence of other components—bonded nitrogen, calcium carbonate, traces of the organic silicon and silicates for the sample machined in oil and an amino group with a carbon atom, bonded in TiN nitrogen, also calcium carbonate for the sample machined in water—can be explained by the chemical composition of the working mediums. The first one correlates to the chemical composition of the organic mineral oil, while the second one correlates to the chemical composition of distillate and deionized technical water.

### 3.6. Discharge Gap

The optically measured cut width $l$ is $0.380 \div 0.400$ mm (Figure 13), the measured distance between the machined surfaces $l_m$ is $1.858 \div 1.978$ mm, and the accuracy of measuring is 5 μm. The programmed distance between two cuts $l_p$ is 2 mm. The calculated discharge gap $\Delta_{DB}$ is in the range of $0.065 \div 0.075$ mm. The recommended offset is $0.190 \div 0.200$ mm, which is larger than for structural anti-corrosion steels (0.175 mm) but less than recommended for aluminum alloys ($0.204 \div 0.207$ mm).

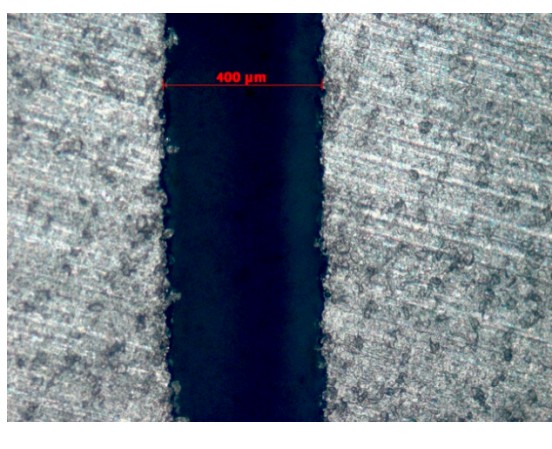
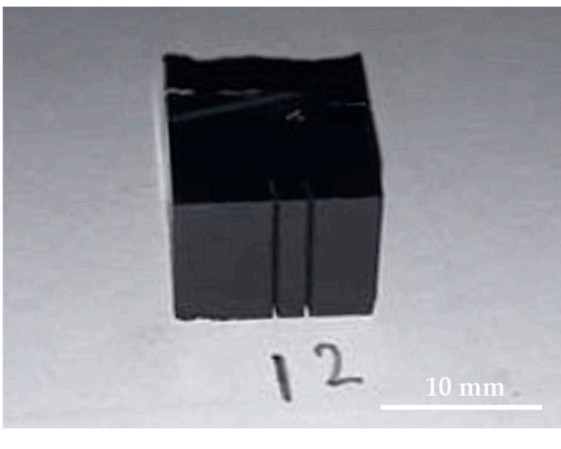

(**a**)        (**b**)

**Figure 13.** Cut slot in the $Al_2O_3 + 30\%TiC$ nanocomposite sample after electrical discharge machining in water: (**a**) microscope picture, 50.0×; (**b**) general view.

## 4. Discussion

As it can be seen, one can estimate the efficiency of electrical discharge machining of nanocomposites by controlling the effective values of the vibroacoustic signal and discharge current. For example, it is known that the size of the well formed after the discharge depends in an extreme way on the pulse duration [33,35,80,81]. It is possible to select the optimal value of the effective values ratio of the signals and current automatically by varying the pulse duration during electrical discharge machining and monitoring the parameters of the recorded signals. Other parameters of the process can be varied in a similar way, including the value of the discharge, by maintaining the constancy of the ratio of useful energy to consumed energy [36,82,83]. It can be seen that, in the latter case, the signal becomes more complex during electrical discharge machining of the nanocomposites compared with other processing methods [7,23]. Quite weighty components appear around the spectral maximum at the frequency of the impulse action, indicating the presence of discharges between the electrodes and non-periodic short-circuits, which contribute to the energy of signals. In this situation, finer filtering of the signal, tuned to the frequency of the applied discharge pulses, may be required.

The microstructure and chemical analyses of the machined nanocomposites (Figures 9 and 10) showed an oxygen unsaturated type of surface destruction. This type of destruction is correlated to the complex wear in the presence of heat when the metastable secondary structure of the second order (formed ceramic plaques with the width of 25–30 μm for the sample machined in water and non-homogenous plaques of 8–10 μm in oil) is adherent to the heat-affected surface of the base nanocomposte. The formed surface and subsurface layers under this type of wear have the presence of partial removal of the secondary structures, which are more brittle in the case of machining in water (Figure 14) [78,79,84–88].

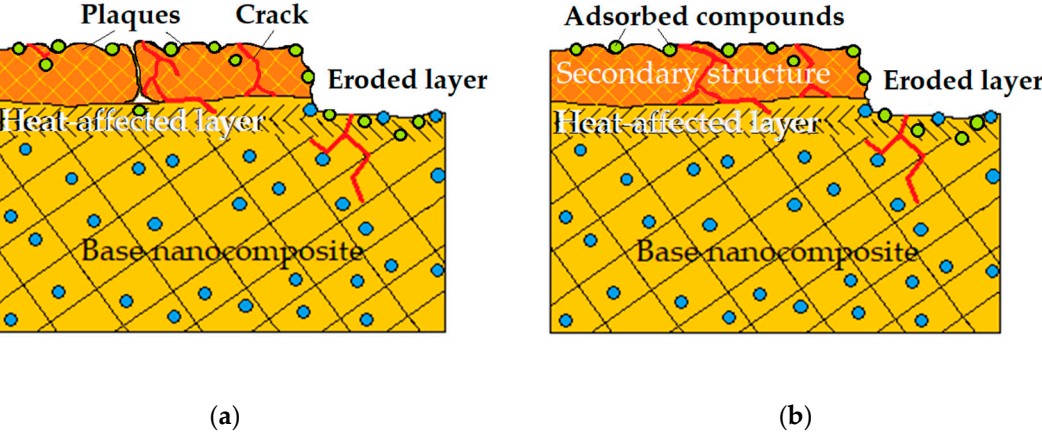

(**a**)             (**b**)

**Figure 14.** The submicron structure of surface and subsurface layers under the oxygen unsaturated type of destruction for ceramic nanocomposite: (**a**) in oil; (**b**) in water.

In the context, zinc oxide (ZnO) formed during machining in oil is an n-type semiconductor and sublimates at a temperature of 1800 °C [89,90]; copper (II) oxide (CuO) that is observed after machining in water does not react with an aqueous medium and decomposes to metallic copper in the presence of hydrogen, carbon, or carbon monoxide [91,92]:

$$CuO + H_2 \rightarrow Cu + H_2O, \tag{3}$$

$$2CuO + C \rightarrow 2Cu + CO_2 \uparrow . \tag{4}$$

This explains the presence of CuO in the sample machined in the water when the metallic copper is observed in the sample produced in oil, where the reaction (4) has more probability due to the prevalence of carbon in the hydrocarbon oil.

At the same time, the possibly formed zinc carbide $ZnC_2$ is not stable in the presence of water and reacts with it to form acetylene $C_2H_2\uparrow$.

Copper (II) carbonate or copper (II) acetanilide is not possible to obtain by the direct exchange reaction in water. Moreover, these compounds are explosive and have particular colors and their presence can be easily detected.

The presence of carbon in the subsurface layer, which is at least 3–5 times higher than in the primary material, cannot be explained by the presence of a heat-affected zone as it can usually be characterized by the low carbon microstructure layer and the following transition zone to the base material [93,94].

However, the mechanism of sublimation for composites and nanocomposites can explain it by the adsorption of formed chemical components of the first order (graphite, metallic copper for the sample machined in oil; metallic titanium and zinc—in water) and the second order (carbides, zinc oxide, silicon compounds, $TiO_2$, $TiN$—in oil; $TiO_2$, $TiN$, $CuO$—in water) by an eroded refractory matrix of base nanocomposite [95–98].

The presence of the significant content of aluminum and carbides in the sample machined in oil (aluminum is 3.8 times higher when there is around 12% of carbides) can prove the formation of aluminum carbide. As it is known, aluminum demonstrates the high chemical reactivity towards oxygen that is higher than for halogens (except fluorine) and with no-metals, including carbon, in the presence of heat. In the proposed conditions of electrical erosion that corresponds to the oxygen unsaturated type of destruction in the presence of heat, when the quantity of oxygen stays less than it should be for alumina forming, aluminum forms pyrotechnical aluminum carbide. It is resistant up to 1400 °C but reacts with water (×12), forming a gelatinous sediment of aluminum hydroxide (×4) and methane (×3):

$$4Al + 3C \rightarrow Al_4C_3, \tag{5}$$

$$2Al_2O_3 + 9C \xrightarrow{1800\ °C} Al_4C_3 + 6CO \uparrow, \tag{6}$$

$$Al_4C_3 + 12H_2O \rightarrow 4Al(OH)_3 \downarrow + 3CH_4 \uparrow . \tag{7}$$

This pyrotechnical aluminum carbide is often called "pyrolytic carbon." However, the correct form probably will be pyrotechnical or pyrolytic aluminum carbide.

Furthermore, the actively formed $Al_3C_4$ is a dielectric that can hamper electroerosive machining by changing the discharge gap's electrical conditions since hydrocarbon oil has better conductivity ($1.9 \div 4.3$ $\Omega\cdot$m depending on oil purity) than isolating ceramics ($\leq 10^{-4}$ $\Omega\cdot$m). However, the discovered proof of formed $Al_4C_3$ has an indirect character despite the observed abundant black sediment during machining and indirect spectroscopy evidence. It means that most of the formed sediment was settled in the medium that was not collected during experiments but can be a direction for further research to unveil wholly and finally the subject of electrical discharge machining alumina and alumina-based materials in oil.

It should be noted that amphoteric oxides—$Al_2O_3$, $TiO_2$, $ZnO$ [99,100]—do not react with water but can interact between each other with the formation of more complex structures, such as insoluble solid solutions: aluminum titanate or tialite $Al_2TiO_5$ (Figure 10), zinc dialuminate (III) gahnite $Al_2ZnO_4$, zinc titanate ($ZnTiO_3$).

## 5. Conclusions

The article presents the first comprehensive and exhaustive study of the process of electrical discharge machining of the ceramic nanocomposite in two working fluids (water and mineral oil) with the developed electrical discharge factors, estimation of the wire tool behavior under pulses, evaluation of the tool material and medium influence and chemical changes that occur in surface and subsurface layers, and estimation of the discharge gap. The principal ability of fine quality surface formation with the Ra of 1.87 µm in water by a single pass of wire tool and precision cut for ceramic

nanocomposite $Al_2O_3$ + 30%TiC was demonstrated when the roughness of the samples produced in oil stays under required level (Ra of 4 μm).

Monitoring of the signals showed that it is possible to solve a number of applied problems of changing the parameters of signals and there is a prospect of creating a multiparameter system for controlling the processes of electrical discharge processing using control of vibroacoustic signals and discharge current, which is more effective in comparison with similar systems based only on measuring electrical parameters that showed their ineffectiveness in the case of machining in oil.

Chemical analyses of the surface and subsurface layers of the samples showed nanomodification by adsorption of newly created components in the nanoframe of the heat-resistant matrix, pellets of solid solutions and plaques of the secondary order under conditions of low-temperature plasma, and the presence of components of the working medium and wire tool at a depth of up to 2.5–4 μm. The compositions are different for each working fluid and following the chemical interactions between elements that occurred in the presence of plasma heat and pressure.

It should be noted that the surface and subsurface layers of $Al_2O_3$ + 30%TiC machined in water contain 3.8 times more aluminum, 8.5 times more titanium compounds (TiN), 1.35 times more traces of atmospheric carbon contaminations, and 1.5 times more oxides (probably partly involved in the bonds with aluminum and titanium). In addition, these samples can be characterized by finer aluminum oxide film (up to 1.5–2.0 μm) that has a presence of plaques (width of 25–30 μm) on the main eroded material and carbon saturated sublayer, with the thickness of 0.5–1.5 μm.

At the same time, the nanocomposite samples machined in oil contain the presence of carbon bonding in graphite and carbides, bonded nitrogen, calcium carbonate, traces of the organic silicon, and silicates that correspond to the chemical content of the hydrocarbon oil. The reduced volume of aluminum can be consumed on the formation of pyrotechnical aluminum carbide from alumina (×2) and carbon (×9) at 1800 °C with CO↑ (×6). The aluminum oxide film is thicker (3.5–5.0 μm) but has a non-homogeneous microstructure (the plaque of 8–10 μm in width) when the carbon saturated sublayer is pronounced and of about 3–5 μm.

The obtained data allows conclusions to be drawn regarding nanomodification of the surface and subsurface layer in the case of oxide nanocomposite machining by using different components of the machining processes and controlling it by choosing the working fluid and material of the electrode tool. It can promote the formation of a fragile sublayer for easier mechanical or electrochemical removal of the products where the recast layer is undesired or, on the contrary, the formation of a stronger sublayer to improve wear resistance of the responsible working surfaces of the product.

The obtained knowledge has a fundamental character and can be used as a recommendation for industrial applications, in terms of the more proper choice of the electrode tool material for electrical discharge machining design for developing technology using newly created conductive ceramic nanocomposites, in the context not only of structural requirements addressed to the working and auxiliary surfaces of the final product, but also of functionality in the exploitation conditions.

## 6. Patents

1. Kozochkin, M.P.; Grigoriev, S.N.; Porvatov, A.N., Okunkova, A.A. The method of controlling the electrical discharge machining of parts on an automated cutting machine with a system of CNC; RU 2598022.
2. Kozochkin, M.P.; Khoteenkov, K.E.; Porvatov, A.N., Grigoriev, S.N. The method of EDM cutting of products; RU 2638607.
3. Grigoriev, S.N.; Kozochkin, M.P.; Okunkova, A.A. The method of positioning the wire electrode on the EDM cutting machines; RU 2572678.

**Author Contributions:** Conceptualization, S.N.G.; methodology, M.P.K.; software, K.H.; validation, P.M.P., A.N.P.; formal analysis, S.V.F.; investigation, M.P.K.; resources, P.M.P., S.V.F.; data curation, P.A.P., K.H.; writing—original draft preparation, A.N.P.; writing—review and editing, A.A.O.; visualization, P.A.P., A.A.O.; supervision, M.A.V.; project administration, M.A.V.; funding acquisition, S.N.G. All authors have read and agreed to the published version of the manuscript.

**Funding:** This work was supported by the Russian Science Foundation under grant 18-19-00599.

**Acknowledgments:** The research was done at the Department of High-Efficiency Processing Technologies of Moscow State University of Technology STANKIN.

**Conflicts of Interest:** The authors declare no conflict of interest. The funders had no role in the design of the study; in the collection, analyses, or interpretation of data; in the writing of the manuscript, or in the decision to publish the results.

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
