# Peer review of "Electrical Discharge Machining of Oxide Nanocomposite: Nanomodification of Surface and Subsurface Layers"

_jmmp, doi:10.3390/jmmp4030096_

Round 1

Reviewer 1 Report

The following comments and remarks should be taken into account in the revision of the manuscript:

1) Line 23 - there is "X-ray" two times

2) Section "Materials and Discharge Gap" should at first in "Materials and Methods" section. "Equipment" and " Characterization of the Samples" should be conected to each other because both describe methods.

3) Line 161 - the manufacturers of Al2O3 and TiC should be mentioned

4) Line 217 - how long was the washing in acetone and how many times. Please describe the washing process.

Author Response

Thank you very much for your review comments. Please find our response at the attachment.

Reviewer 2 Report

The present work investigates materials changes due to electrical discharge machining of conductive nanocomposite based on Alumina. The work is interesting, a lot of data were collected and discussed. Nevertheless I strongly suggest minor revision before publishing. 

  • Quality of Figure 1 and Figure 2 and Figure 9 must be improved. 
  • It must avoided to divide a table among different pages such as Table 2, Figure 9 and Table 4.

Author Response

(The authors gave the same response as above.)

Reviewer 3 Report

The Electrical Discharge Machining of Oxide Nanocomposite: Nanomodification of Surface and Subsurface Layer

In this paper the authors have presented their work on the electrical discharge machining (EDM) of conductive Alumina based nanocomposites using a brass tool. The nanocomposite of Al2O3 70% + 30% of TiC was electroerosively machined in a water and hydrocarbon oil. The machined surface of the samples were characterized using scanning electron microscopy and X-ray photoelectron spectroscopy. The surface and subsurface analysis showed process-dependent variation in the modification of surface/subsurface. According to the authors the nanomodification occurs by adsorption of newly created components of the ceramic composites and secondary phases under conditions of low-temperature plasma of EDM process and the presence of working medium and wire tool. The surface chemical compositions were different for each working fluid because of differences in the  chemical interactions between fluids, elements and the presence of plasma, heat and pressure.

Comments:

  1. This paper is a comprehensive research with introduction, experimental details, results, discussion, and conclusions. The reference list is recent and thorough.
  2. English language fine tuning such as proper sentence structure is needed in several places.
  3. Line 242-243, why surface topology of surface after machining in water has more even structure as compared to that with machining in oil?
  4. In Fig 6 and Fig 7 why chemical composition line profiles across sample (in vertical direction) were not acquired along with elemental mapping?
  5. Line 261-263: "The titanium and carbon depletion can be also observed (Figure 7d,e) as a presence of solid solution of aluminum and titanium and more complex compounds in the form of pellets (Figure 7b,d). What is the length-scale of these pallets? Is it possible for authors to take top (plan) view image using SEM and prove this?
  6. Fig 8. It is not very clear for reader to know which geometry authors are talking about and which element is represented by particular color. Can Authors provide more clarity to this figure?
  7. Table 4. The atomic % values are calculated by curve fitting XPS curves. What are % errors (random measurement) in Atomic % values?

Author Response

(The authors gave the same response as above.)
